# Assessment of the Medical Reliability of Videos on Social Media: Detailed Analysis of the Quality and Usability of Four Social Media Platforms (Facebook, Instagram, Twitter, and YouTube)

**DOI:** 10.3390/healthcare10101836

**Published:** 2022-09-22

**Authors:** Deniz Gurler, Ismail Buyukceran

**Affiliations:** 1Department of Orthopedics and Traumatology, University of Health Sciences, Samsun Training and Research Hospital, Samsun 55090, Turkey; 2Department of Orthopedics and Traumatology, Faculty of Medicine, Ondokuz Mayıs University, Samsun 55139, Turkey

**Keywords:** social media, YouTube, Instagram, Twitter, Facebook, video, knee replacement

## Abstract

*Introduction:* In recent years, the internet and social media have become the primary source for patients to research their medical conditions. Given the billions of links that result from research, it has become increasingly important how medically high quality the priorities of the search algorithms are. This study aims to examine the medical quality of videos on social media. *Material and Method:* A new Gmail account was never used, and Facebook, Instagram, Twitter, and YouTube accounts were opened. The word “knee replacement” was searched via social media. The video duration, daily views, total views, number of likes, source, and shared content were recorded. The parameters were statistically evaluated using the scales JAMA, GQS, DISCERN, and VPI to measure the quality of the medical posts. *Results:* Correlations were found between JAMA, GQS, and DISCERN. No correlation between the VPI scales with other scales was found. It was found that the promotional content in videos other than Instagram was very high (56–70%). Academics and healthcare workers produced greater quality content than other groups. There is a clear dominance of healthcare practitioners on Instagram. The most shared content was informative, depending on the content. The most frequent users were alternative health practitioners. While YouTube had the highest JAMA, GQC, and DISCERN (2.98, 3.18, 37.5) scores, the lowest VPI (0.761) score was found. *Discussion and Conclusions:* It has been shown that Instagram and Twitter are not the right places to share videos with medical content. However, everyone should remember that Instagram is the best place to share short but popular videos. YouTube and Facebook are available resources to share videos of better medical quality with a higher score than others. We can say that the most reliable medical sources are Facebook and YouTube videos of physicians and medical staff.

## 1. Introduction

The internet undeniably occupies a vital place in our lives. Today, in many areas, it is impossible to imagine life without the internet in our family, work, and social life. Health is not an area exempt from the internet either. The internet has become the classic approach to gathering information about their illness. We increasingly observe that patients watch videos of recommended surgeries. It is also known that 77% of patients search for hospitals in search engines before visiting them, and 33% of them visit hospital websites [1]. In this case, a significant problem arises. The internet can sometimes be full of junk information and misinformation. It is a crucial problem to find medically accurate information on the internet. Of course, this topic has already been found worthy of investigation [2]. Videos are the most influential media on the internet. Based on human evolutionary history, visual materials have a more effective and lasting impact. Research specific to education shows that audio and visual materials are more effective than verbal materials. Cognitive effects of educational approaches supported by images are also known [3,4,5]. When talking about videos in the internet universe, the first website is undoubtedly YouTube. However, social media are other internet resources used more frequently than YouTube [6]. Although YouTube is the only source in almost all previous studies, other social media and videos should also be considered. Therefore, one of the two main approaches of this study is to reach out and compare as many different social media videos as possible. The second approach of the study is to evaluate the medical quality of videos about a standard surgical procedure. Therefore, the topic chosen for the study was “knee arthroplasty”, one of the most commonly performed surgeries in orthopedic clinics. 

## 2. Materials and Methods

A local ethics committee (Samsun University Clinical Research Ethical Committee) was contacted for an ethical review of the study. The study was approved with the decision number SUKAEK 2022/5/8. The study was designed to investigate the medical accuracy of the information obtained from internet searches on a medical topic. The study was prospective in design. The criteria were as follows. The new Gmail account “kneerplcmnt@gmail.com” and the same mail account associated with Facebook, Instagram, Twitter, and YouTube accounts were opened to obtain data independent of searching algorithms. Google Chrome, the most widely used browser among mobile browsers and PCs, was used for the search [7]. First, Chrome was installed with a Gmail account on a PC (Asus ZenBook UX305UA) where no other version was registered, and the search was performed using this browser. Since the PC and browser versions of Instagram do not fully support the features of the Instagram video section, an android PC emulator (BlueStacks 5.8.101.1001 N32) was used. The browser search was completed on 26 July 2022, in Samsun, Turkey, localization (41°16′18.7″ N 36°17′53.4″ E) via Turkcell Superonline (100 Mbps fiber optic internet) using a PC with Windows 10 Home (21H2) installed. Three phrases (“knee replacement”, “kneereplacement”, and “#kneereplacement”) were searched in the general search bars of the social media accounts. Videos in non-English languages and reposted videos were excluded. Previously, videos were excluded from some studies for their low number of likes, views, and duration. Therefore, such a distinction is wrong for a comprehensive comparison of video algorithms because it implies that the researcher manipulates the study group data. Thus, the short duration and a low number of likes and views were not used as exclusion criteria to prevent data manipulation.

The first 50 videos from the search results on all four social media platforms were evaluated for the study. In determining this number, it was seen that 90% of YouTube users stop watching after the first 30 videos. Since four different platforms were used, the number of samples was fixed at 50 [8,9,10,11,12,13]. Several variables were recorded, including the duration of the videos, the time of the first post share, the JAMA Criteria [14], GQS (Global Quality Score for Educational Value) [15], VPI (Video Power Index) [16], DISCERN (Quality Criteria for Consumer Health Information) [17,18], and the content, occupation, and location of the participants (account owners, academics, physicians, non-physicians, patients). JAMA, GQS, and DISCERN criteria are listed in the table (Table 1). The aim was to measure both the popularity and the medical quality of the video by using scales recommended for medical reasons as well as measuring the popularity of the video. Of these four scales, VPI and GQS have already been used for various purposes in non-scientific settings. Still, JAMA and DISCERN are essential scales developed for the online quality of medical articles. For this reason, there are few publications in which these four scales are applied and used simultaneously. However, there is no study in which they are all used, and videos from four different platforms are compared. 

The original VPI formula was (number of likes/dislikes + likes) × 100, but after November 11, 2021, the numbers on the YouTube dislike number were hidden. Therefore, a modified formula was used: (number of likes/number of views) × 100. This formula was also used for YouTube videos in this study [10,13].

The academic group was associated with a university or medical research group. Physicians were not affiliated with a university or a research center. Non-physician groups were health professionals and physical therapists. Alternative groups were non-health worker trainers, massage therapists, healers, and alternative medicine providers. Although alternative medicine is usually included in the non-physician group in many studies, the need for a separate grouping has arisen from the fact that it is very prominent in Instagram videos. The patient group consisted of patients who had undergone previous knee surgery. The sharing contents (information about the disease, training, exercise, surgical technique, patient experiences, and advertising) were recorded. Those who reference a profit making entity or organization in their posts or account information (even if it was a health care entity) have been registered in the ad group. The study also examined the number of daily views (total views/total online days), which was not previously defined or used in the studies. Unlike other studies, this parameter focuses on popularity and usability among internet users, regardless of the standard sites, as a criterion for regular video access.

### Statistical Analysis

The data files were processed using Excel 2016. Then, analyses were performed using the statistical analysis program Jamovi 2.3.13 (the Jamovi Project, Sydney, Australia). The relationship between the data was studied using social media sites (Facebook, Instagram, Twitter, and YouTube). Values (popularity and medical information) were compared by the source of publication and social media. Mann–Whitney U was used for the non-normal distribution. Spearman correlation tests were used for the relationships between parameters.

## 3. Results

Of the 264 videos searched on four social media outlets, the first 200 (4 × 50) videos that met the criteria were included. When the descriptive averages of all videos are combined, the table is remarkable (Table 2). Percentages of content and participants by account and job type were examined (Figure 1 and Figure 2). It was deemed appropriate to present four different social media outcomes discussed in the Results section separately. 

### 3.1. Facebook Results

The examination of Facebook videos reveals an average duration of 230 seconds (±275), daily views 594 (±2071), JAMA 2.3 (±0.58), GQS 1.66 (±0.717), views 641,587 (±2.99e+6), VPI 2.45 (±2.39), DISCERN 23.1 (±6.44). Source distribution was academic 0%, physician 70%, non-physician 18%, patient 12%, alternative medicine 0%. According to the sharing source, the distribution was as follows; information 34%, training 10%, surgery 4%, patient experience 52%, and advertising 68%. A total of 73 videos were viewed. twenty videos from languages other than English were excluded from the study. Three videos were shared twice.

There was a moderate correlation between JAMA criteria GQS (ρ = 0.500, *p* < 0.001) and DISCERN (ρ = 0.550, *p* < 0.001) in Facebook videos. There was a high correlation between GQS and DISCERN (ρ = 0.779, *p* < 0.001). Only a moderate correlation was found between the number of daily views and VPI (ρ = 0.474, *p* < 0.001). When compared with source, JAMA (X^2^ = 6.75, *p* = 0.034), GQS (X^2^ = 5.77, *p* = 0.050), and duration (X^2^ = 10.57, *p* = 0.005), a significant difference was found. JAMA scores were significantly higher in physician and non-physician posts than in patients’ posts (W = −3.59, *p* = 0.030). Non-physician posts had higher GQS values than others (W = 3.327, *p* = 0.049). Non-physician participants shared longer videos than others (W = 4.28, *p* = 0.007). Content, JAMA (X^2^ = 10.26, *p* = 0.016), GQS (X^2^ = 14.92, *p* = 0.002), DISCERN (X^2^ = 14.28, *p* = 0.003), duration (X^2^ = 7.95, *p* = 0.047), were found to have significant differences when compared. For patient experience, the scores for JAMA (W = −4.771, *p* = 0.004), DISCERN (W = −4.156, *p* = 0.017), and GQS (W = −4.656, *p* = 0.005) were significantly lower, and the training videos had higher scores.

### 3.2. Instagram Results

When a total of 80 Instagram videos were examined, the following results were obtained: Average duration of 33.3 seconds (±20.8), daily views 4487 (±10,775), JAMA 1.66 (±0.519), GQS 1.38 (±0.567), views 47,006 (±66008), CPI 5.29 (±4.19), DISCERN 18.7 (±3.42). The distribution according to the source was as follows; academic 2%, physician 14%, non-physician 6%, patient 34%, alternative medicine 44%. By content, informational 14%, training 46%, surgical 0%, patient experience 40%, advertising 6%. The difference between Instagram and other social media was that there was much reposting, and many posts were displayed in a foreign language. The number of reposted videos was 17, and the number of non-English videos was 13.

When viewing Instagram videos, there was a low correlation between the JAMA criteria and DISCERN (ρ = 0.292, *p* = 0.040). There was a moderate to the high correlation between DISCERN and GQS (ρ = 0.781, *p* < 0.001). There was no correlation between the number of daily views and all video quality ratings. Depending on the source, there was a significant difference between daily viewing (X^2^ = 15.68, *p* = 0.003), GQS (X^2^ = 9.96, *p* = 0.041), DISCERN (X^2^ = 10.56, *p* = 0.032). Alternative medicine posts received more daily views than others (W = 5.127, *p* = 0.003). Academic and physician posts had higher GQS values than others (W = −3.571, *p* = 0.05). Posts about alternative medicine seemed to receive more likes than the others (W = 6.449, *p* < 0.001). According to the content, there was a significant difference in daily view (X^2^ = 12.710, *p* = 0.002), GQS (X^2^ = 12.778, *p* = 0.002), DISCERN (X^2^ = 15.102, *p* < 0.001). It was clear that the training posts received more daily views and GQS values than the others (W = −5.236, *p* < 0.001, W = −4.666, *p* = 0.003). It was found that the training videos received more likes and higher DISCERN scores than the others (W = −6.48, *p* < 0.001, W = 5.0987, *p* < 0.001).

### 3.3. Twitter Results

When analyzing 61 Twitter videos, the average duration is 36.3 seconds (±30.8), daily views 1.17 (±2.45), JAMA 2.36 (±0.693), GQS 1.58 (±0.758), views 234 (±340), CPI 6.23 (±5.29), DISCERN 18.8 (±3.28). Because it is possible to link to other video sources (YouTube, et cetera) on Twitter, eight posts that contained external links were excluded from the study. Three reposted videos were excluded.

The distribution according to the source was as follows; academic 2%, physician 30%, non-physician 46%, patient 22%, alternative medicine 0%. Content, by information 40%, training 4%, surgery 8%, patient experience 48%, advertising 56%. 

When examining the videos in the Twitter search results, there was a moderate correlation between the GQS of the JAMA criteria (ρ = 0.556, *p* < 0.001) and DISCERN (ρ = 0.489, *p* < 0.001). There was a high correlation between GQS and DISCERN (ρ = 0.665, *p* < 0.001). There was only a moderate correlation between daily visual counts and VPI (ρ = 0.403, *p* = 0.004) and JAMA (ρ = 0.556, *p* < 0.001). Depending on the source, a significant difference was found between JAMA (X^2^ = 8.15, *p* = 0.043), GQS (X^2^ = 13.51, *p* = 0.004), and DISCERN (X^2^ = 12.34, *p* = 0.006). The physician appeared to have higher JAMA scores than the others (W = −3.315, *p* = 0.050). Academics and physicians had higher GQS (W = −4.69, *p* = 0.005) and DISCERN (W = −4.41, *p* = 0.010) scores than others. Regarding content, JAMA (X^2^ = 6.861, *p* = 0.050) and GQS (X^2^ = 8.721, *p* = 0.033) differed significantly. It was clear that sharing patient experiences had lower JAMA and GQS scores than others (W = −3.204, *p* < 0.001, W = −3.442, *p* = 0.050).

### 3.4. YouTube Results

Based on 50 YouTube videos, the average duration was 859 seconds (±1103), daily views 280 (±750), JAMA 2.98 (±0.958), GQS 3.18 (±1.06), views 322,732 (±536,264), VPI 0.761 (±0.54), DISCERN 37.5 (±11.4). The distribution according to the source was as follows; academic 6%, physician 68%, non-physician 18%, patient 8%, alternative medicine 0%. Content, by information 56%, Training 12%, surgery 24%, patient experience 8%, advertising 70%. 

YouTube videos showed a high correlation between JAMA criteria GQS (ρ = 0.743, *p* < 0.001) and DISCERN (ρ = 0.638, *p* < 0.001). A high correlation was observed between GQS and DISCERN (ρ = 0.800, *p* < 0.001). A high correlation was observed between daily viewership and JAMA (ρ = 0.743, *p* < 0.001). Depending on the source, a significant difference was observed between JAMA (X^2^ = 93.90, *p* = 0.025) and DISCERN (X^2^ = 13.195, *p* = 0.004). Posts from physicians were found to have higher JAMA and DISCERN scores than others (W = −3.886, *p* = 0.031, W = −3.99, *p* = 0.025). It can be seen that posts from physicians receive many more likes than others (W = −3.7659, *p* = 0.039). JAMA scores (X^2^ = 9.24, *p* = 0.026) significantly differed when considering the content. Surgical video posts had higher daily views and GQS values than others (W = 2.8818, *p* < 0.050). Informational videos had higher JAMA scores than others (W = −4.154, *p* = 0.017). Patient experience videos had lower DISCERN values (W = −2.813, *p* = 0.019). It was clear that surgical videos received more likes than others (W = 3.522, *p* = 0.05).

Overall, there were significant moderate to strong correlations between JAMA, GQC, and DISCERN. On the other hand, it was found that there is no overall correlation between the different scales in the VPI. In summary, it is possible to examine these results using the cross-correlation table with JAMA, GQC, VPI, and DISCERN (Table 3).

## 4. Discussion

A basic unfiltered Google search yields 3.9 million video links for “knee replacement” [20]. It is impossible not to lose oneself in this vast number. For this reason, reaching for the correct information and reliable posts is much more critical. Website and social media search algorithms occupy an essential position. However, social media algorithms are issues that change and evolve much faster.

All social media sites use different algorithms. The order in which the posts are displayed to the user is a critical point. The primary purpose is based on the concept of personalized advertising to the customer proposed by the advertising industry. Therefore, most social media algorithms are almost exclusively about personalization. The algorithms used by the social media accounts that are the subject of our study are complex systems that are frequently modified and commercially confidential. 

Facebook completely renewed its algorithm after 2019 [21]. It has started to display original, high-quality videos that are longer than three minutes and are viewed for more than one minute. The new algorithm corresponds to a study conducted by Yurdaisik in 2020, and it was found to be consistent with the average of 4.47 min [16]. When a very specific search term (e.g., glioblastoma) is chosen, as in the 2018 study by ReFaey et al., very nonspecific results can also be obtained. In their study, 88% of the results were hospital or university sharing [22]. The foundation of the algorithm is based on a few main pillars. Who shares Facebook posts and content, the type, and interactions are critical to the algorithm. For Facebook videos, patient experience is rated very low in JAMA, GQC, and DISCERN; in contrast, physicians received higher ratings than the others. Educational and training videos received higher ratings than others. In the study by Szumada et al. in 2020, 30.7% of educational videos represent a very important proportion [9]. The results support this for Facebook, which uses an algorithm where content matters.

Contrastingly, Instagram’s algorithm is quite different. The key elements are the user’s relationships, interests, and relevance [23]. The user’s relationship with other users on Instagram is more critical than on other social media. For this reason, the algorithm does not allow sharing for advertising purposes. Since business and official accounts are registered differently on Instagram, advertisement images are rare when searching Instagram Reels. The low percentage of ads in Instagram videos (6% in our study) proves this. However, this discrepancy is also consistent with the abundance of healers and alternative medicine providers on Instagram, which we can see as hidden advertising accounts. The limited duration of Instagram videos shows that it is a different area for sharing videos than other social media. It is impossible to share videos longer than 90 s on Instagram. Although this is a drawback, VPI ratings were 6.6 times higher than YouTube videos and 2.1 times higher than Facebook videos. This result suggests that VPI ratings are not very successful in indicating the strength and quality of videos. The popularity of videos that are very short and of low medical quality can be very high. Similarly, Instagram was found to have the highest daily views on social media. Instagram Reels videos have 7.6 times more daily views than Facebook videos and 16 times more than YouTube videos.

Twitter uses an algorithm first to suggest general trending topics. However, it also tries to display personalized trending topics at the top of the rankings. Timeliness, relevance, engagement, and rich media content are the other main elements of Twitter’s algorithm. Twitter’s trending topic algorithm determines which topics are displayed as trending. Twitter’s data shows that video views on Twitter have increased by 95% in 18 months, and 71% of Twitter sessions now include videos [24]. Despite the brevity of video duration on Twitter, the highest VPI values in our study (6.23) confirm this. The academic and physician groups scored significantly higher in JAMA, GQS, and DISCERN scores. Patient experience was observed to be significantly lower on the medical quality scales.

Personalization, video performance, and external factors (especially relevance) are the main parameters for YouTube video ranking [25]. The lowest average VPI values (0.8) of the YouTube videos in the study suggest that the video performance used in the algorithm is inconsistent with the VPI. It is significant that YouTube videos have high JAMA, GQC, and DISCERN values. These scales, which have a significantly higher correlation than other areas of social media, have demonstrated the usability and meaningfulness of YouTube in medical posts. 

When all groups were examined, the proportion of academics and physicians was found to have higher quality medical content. The abundance of videos for explicit or covert advertising purposes is striking. Aside from Instagram, which prevents sharing due to its algorithm, 50–70% of other sites openly share for advertising purposes. This high rate is far from being an area created for people trying to use and share social media to obtain accurate information. The quality of advertising content related to medical information is relatively low.

The rating of the videos by one person is an important deficiency, and it would be better to have the scales rated by one or more people for control purposes. In the study, the comparison of videos from four social media platforms, which has not been carried out before, is seen as both an advantage and a disadvantage. While the comparison of social media videos that address very different areas is a major handicap, the limitations in standardizing the groups ensured the scientific similarity of the compared groups. The study is valuable because it makes comparisons between videos on different platforms, which are very difficult to compare.

## 5. Conclusions

The value of video, the most effective method of using visual and audio materials, in education and communication is undisputed. When it comes to sharing videos online, YouTube is undoubtedly the first source that comes to mind, while all social media platforms provide options to their users by creating video-based algorithms. The purpose of this study is to investigate the medical quality of videos on the most popular social media platforms. Although the data obtained from this study are quite diverse, some important themes have attracted attention.

There is a very serious negative relationship between the popularity of videos and their medical quality. This is not only true for videos with a medical background. If we look at the whole video universe, it is known that educational and informational videos are not very popular. Nevertheless, this study provided data that can be used to improve the usability and medical quality of videos included in scan results. While Instagram and Twitter can be reached quickly and appeal to many people with short videos, their medical quality has been shown to be very low. YouTube and Facebook are considered more valuable options from a medical perspective. Considering the success of academic and medical videos, it is clear that these sites are more successful than others. Moreover, a medical professional who wants to create an accurate and good source of information should prefer YouTube or Facebook.

Instagram and Twitter’s algorithms prevent the production and use of medically long-term and usable videos. Although long videos are not allowed, they can be used on Instagram to reach users quickly and obtain many views. However, Twitter is also not suitable for these purposes. It is clear that videos for advertising and promotional purposes are very popular and have many views. Companies that deal with this issue and produce videos for promotional purposes should take this information into account. In summary, it is necessary to increase individual accounts of academics and physicians on all platforms to provide accurate information related to videos and in all areas. Such accounts will allow patients to access more accurate information on social media.

## Figures and Tables

**Figure 1 healthcare-10-01836-f001:**
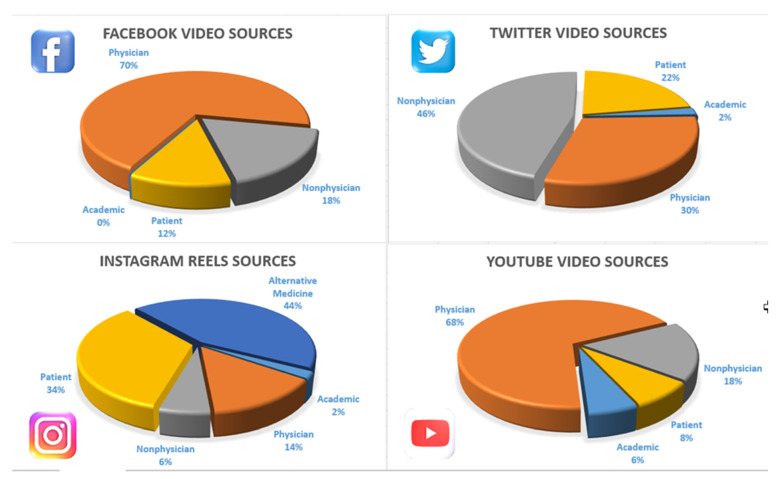
Distribution chart of sources by social media.

**Figure 2 healthcare-10-01836-f002:**
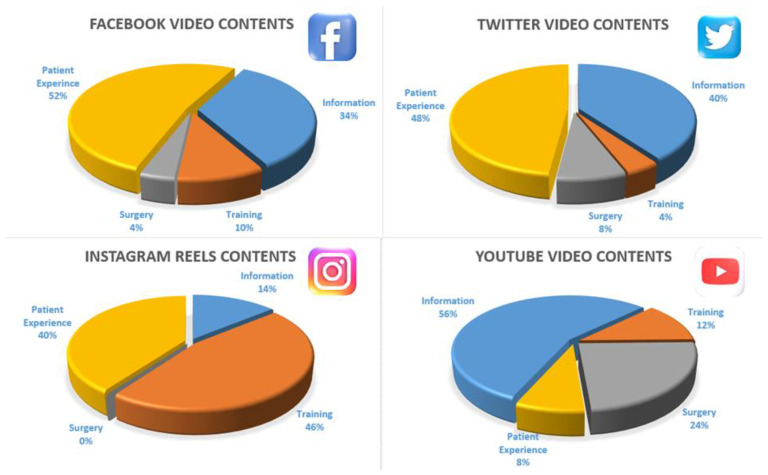
Distribution chart of the content by social media.

**Table 1 healthcare-10-01836-t001:** Criteria for DISCERN, GQC, and JAMA scales.

**DISCERN ^1^ Scoring System**		
Section	Questions	Score (1–5)
Reliability of the publication	Explicit aims	1 2 3 4 5
	Aims achieved	1 2 3 4 5
	Relevance to patients	1 2 3 4 5
	Source of information	1 2 3 4 5
	Currency (date) of information	1 2 3 4 5
	Bias and balance	1 2 3 4 5
	Additional sources of information	1 2 3 4 5
	Reference to areas of uncertainty	1 2 3 4 5
Quality of information on treatment choices	How treatment works	1 2 3 4 5
	Benefits of treatment	1 2 3 4 5
	Risks of treatment	1 2 3 4 5
	No treatment options	1 2 3 4 5
	Quality of life	1 2 3 4 5
	Other treatment options	1 2 3 4 5
	Shared decision making	1 2 3 4 5
Based on the answers to all of these questions, rate the overall quality of the publication as a source of information about treatment choices		1 2 3 4 5
**Global quality scoring**		**Score**
Poor quality, poor flow of the site, most information missing, not at all useful for patients		1
Generally poor quality and poor flow, some information listed but many important topics missing, of very limited use to patients		1
Moderate quality, suboptimal flow, some important information is adequately discussed but others poorly discussed, somewhat useful for patients		1
Good quality and generally good flow, most of the relevant information is listed, but some topics not covered, useful for patients		1
Excellent quality and excellent flow, very useful for patients		1
**JAMA ^2^ scoring system**		**Score**
Authors and contributors, their affiliations, and relevant credentials should be provided	Authorship	1
References and sources for all content should be listed clearly, and all relevant copyright information should be noted	Attribution	1
Website “ownership” should be prominently and fully disclosed, as should any sponsorship, advertising, underwriting, commercial funding arrangements or support, or potential conflicts of interest	Disclosure	1
Dates when content was posted and updated should be indicated	Currency	1

^1^ DISCERN: Quality Criteria for Consumer Health Information. ^2^ JAMA: The Journal of American Medical Association.

**Table 2 healthcare-10-01836-t002:** The averages of all of the videos on social platforms.

	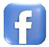 Facebook	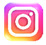 Instagram	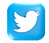 Twitter	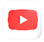 YouTube
Duration Time ^1^	230	33.3	36.3	859
Daily view ^2^	594	4487	1.17	280
JAMA	2.3	1.66	2.36	2.98
GQS	1.66	1.38	1.58	3.18
VPI	2.45	5.29	6.23	0.761
DISCERN	23.1	18.7	18.8	37.5

^1^ In Seconds. ^2^ Formula= total view/(search date-first upload date), social media icons [19].

**Table 3 healthcare-10-01836-t003:** Correlations of the scales on which videos are rated on social media sites.

		JAMA	GQS	DISCERN	VPI
Facebook	JAMA	---	++	++	---
	GQS	---	---	+++	---
	VPI	---	---	---	---
	DISCERN	---	---	---	---
Instagram	JAMA	---	---	+	---
	GQS	---	---	++	---
	VPI	---	---	---	---
	DISCERN	---	---	---	---
Twitter	JAMA	---	++	++	---
	GQS	---	---	---	---
	VPI	---	---	---	---
	DISCERN	---	---	---	---
YouTube	JAMA	---	+++	+++	---
	GQS	---	---	+++	---
	VPI	---	---	---	---
	DISCERN	---	---	---	---

JAMA: The Journal of American Medical Association Benchmark Criteria), GQS: Global Quality Score for Educational Value, DISCERN: Quality Criteria for Consumer Health Information, VPI: Video Power Index. (+: low correlation, ++: medium Correlation, +++: high correlation.).

## Data Availability

The sharing link can access study statistics reports. https://www.dropbox.com/s/5uswescs4j4msqq/contact.denizgurler%40gmail.com.for.pw.rar?dl=0 (accessed on 22 August 2022).

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
