# Peer review of "Assessment of the Medical Reliability of Videos on Social Media: Detailed Analysis of the Quality and Usability of Four Social Media Platforms (Facebook, Instagram, Twitter, and YouTube)"

_healthcare, 2022, doi:10.3390/healthcare10101836_

Round 1
Reviewer 1 Report
Major revision
With the popularity of internet medical, many patients choose to query disease and related content by themselves, but the correctness of the content distributed on the Internet cannot be guaranteed. Therefore, this paper compares the reliability of the four most common social media (Facebook, instagram, twitter, YouTube) in obtaining medical information through videos, and further selects "knee replacement" as the research theme to evaluate the medical quality of videos about a standard surgical procedures.
Introduction
1.“It is common for patients who have received a diagnosis for the first time to pass on the information they have obtained via the Internet to the physicians they go to for their control examinations or a second opinion. Patients often indicate that they watch videos of proposed surgeries”. What is the reference for this sentence?
2.The conclusion of first paragraph is “visual materials have a more effective and lasting impact”,the supporting arguments are seemed too little, please improved.
Materials and Methods
1. Why excluded videos in non-English languages and reposted videos? What is the basis? More forwarding times, more reliable, Isn't it? And how to ensure the representativeness of samples.
2. The sample selection why are the first 50 videos that met the study criteria, rather than sampling, Is there any theoretical basis?
3. State the reference source of the modified VPI formula.
4. Have these four criteria(JAMA Criteria , GQS, VPI , DISCERN) been used as articles to evaluate the reliability of video information in the past?
Results and Discussion
1.The relationship between the data was studied using social media sites (Facebook, Instagram, Twitter, and YouTube).
Conclusions
1.Instagram and Twitter offer very few opportunities for accurate and valuable information. However, the success of academic and medical videos, YouTube and Facebook are considered more valuable options. A person who trying to create an accurate and good source of information ,like a physician or health care, workes hould prefer YouTube or Facebook.
2.No elaborate the shortcomings and advantages of this article.
3.The conclusion of this paper is of weak practical significance.
In addition, few references and english expressions need further improvement.
Reviewer 2 Report
Review of the article: Which social media is more reliable for getting medical information through videos? Detailed analysis of the quality and usability of four social media platforms (Facebook, Instagram, Twitter, and YouTube).*
The idea to study is good. An interesting aspect of the study is usability of social media platforms. However, this article needs some clarification and correction.
Major comments
· In my opinion, the title of the paper could be more exact and informative. I suggest the authors reconsider and correct the title into a more generalized form.
· References are very few.
Chapter - Materials and Methods
Please explain why these scales were used for analyses.
Explain in more detail what each of the scales used in the study examines..
Chapter - Results
Please provide information on how many videos (numbers) were analyzed from each social platform.
Chapter – Discussion
Discuss your results compared to other studies.
Chapter – Conclusions
Conclusions should be redrafted to be based on the results obtained.
Minor comments
Line 12: replace the word “Instruction” with “Introduction”
In Table 2, in addition to the symbols, include the names of social media.
Round 2
Reviewer 1 Report
Adjust the format and serial number of references.
Reviewer 2 Report
The manuscript has been corrected, but requires a slight editorial correction.